# PREDICTING MASKED TOKENS IN STOCHASTIC LOCATIONS IMPROVES MASKED IMAGE MODELING

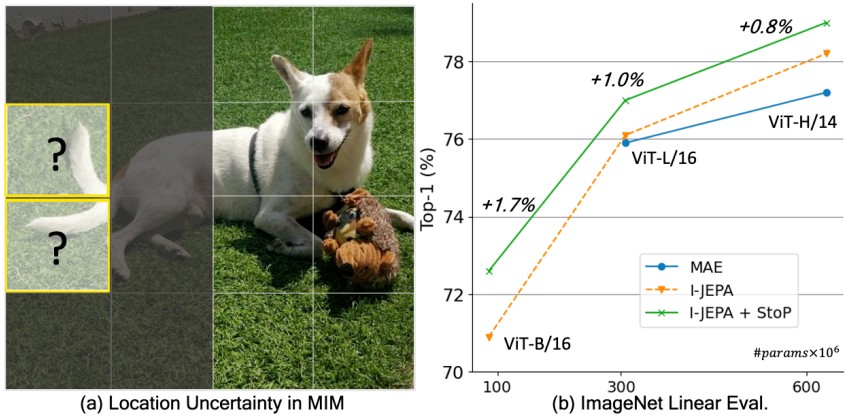

Figure 1: Given a partial image of a dog, can you precisely determine the location of its tail? Existing Masked Image Modeling (MIM) models like MAE (He et al., 2021) and I-JEPA (Assran et al., 2023) predict tokens deterministically and do not model location uncertainties (a), We propose to predict target tokens in stochastic positions (StoP) which prevents overfitting to fixed target locations. Changing existing positional embeddings with StoP leads to improved MIM performance on downstream tasks including linear probing on ImageNet (b).

## ABSTRACT

Masked Image Modeling (MIM) is a promising self-supervised learning approach that enables learning from unlabeled images. Despite its recent success, learning good representations through MIM remains challenging because it requires predicting the right semantic content in accurate locations. For example, given an incomplete picture of a dog, we can guess that there is a tail, but we cannot determine its exact location. In this work, we propose to incorporate location uncertainty to MIM by using stochastic positional embeddings (StoP). Specifically, we condition the model on stochastic masked token positions drawn from a gaussian distribution. We show that using StoP reduces overfitting to location features and guides the model toward learning features that are more robust to location uncertainties. Quantitatively, using StoP improves downstream MIM performance on a variety of downstream tasks. For example, linear probing on ImageNet using ViT-B is improved by $+1.7\%$, and by $2.5\%$ for ViT-H using $1\%$ of the data.

## 1 INTRODUCTION

Masked Image Modeling (MIM) enables learning from unlabeled images by reconstructing masked parts of the image given the rest of the image as context. Recently, new MIM methods have emerged (Xie et al., 2021; Bao et al., 2021; He et al., 2021; Assran et al., 2023). Masked Auto-Encoders (MAE) (He et al., 2021) are trained to minimize a reconstruction error in pixel space, and I-JEPA (Assran et al., 2023) reconstructs image features. MIM is appealing compared to invariance-based self-supervised learning methods like DiNO (Caron et al., 2021) and iBOT (Zhou et al., 2021) as MIM do not suffer from the same limitations, namely it does not require heavy use of hand-crafted augmentations (Xiao et al.; He et al., 2021) or a uniform cluster prior (Assran et al., 2022).

Despite the recent success of MIM, we argue that learning good representations using MIM remains challenging due to location uncertainties because it requires predicting the right semantic content in

accurate locations. For example, given an incomplete picture of a dog (see Figure 1a), we can guess that there is a tail but we cannot determine its exact location as it can naturally appear in multiple plausible locations. Without explicitly modeling this location uncertainty, existing MIM models like MAE and I-JEPA might overfit on semantic content in arbitrary locations.

In this work, we propose to address location uncertainty in MIM by turning existing MIM models into stochastic ones. Instead of training the model to make predictions in exact locations, we propose to use Stochastic Positional embeddings (StoP) to introduce noise to the masked token's positions, implicitly forcing the model to make stochastic predictions. StoP guides the model towards learning features that are more resilient to location uncertainties, such as the fact that a tail exists somewhere in a broad region of the image, and this in turn leads to improved performance (Figure 1b).

Specifically, we model the location of every masked token as a random variable with a Gaussian distribution where its mean is the position of the patch, and the covariance matrix is learned. We find that it is crucial to design StoP carefully so that the model does not merely scale down the covariance matrix weights to overcome the noise and propose to use regularization to alleviate this difficulty.

Our contributions are as follows. First, we propose the idea of Stochastic Positional embeddings (StoP) and apply it to MIM to address the location uncertainty in MIM, namely that the location of semantic features is stochastic. Second, we demonstrate that adding StoP to I-JEPA, a recent MIM approach, leads to improved performance on a variety of downstream tasks, highlighting its effectiveness. Lastly, StoP can be simply plugged into existing models, requiring only three additional lines of code, without adding any runtime or memory overhead.

## 2 RELATED WORK

**Masked image modeling (MIM).** There is a significant body of research exploring visual representation learning by predicting corrupted sensory inputs. Denoising autoencoders (Vincent et al., 2010), for example, use random noise as input corruption, while context encoders (Pathak et al., 2016) regress an entire image region based on its surrounding. The idea behind masked image modeling (He et al., 2021; Xie et al., 2021; Bao et al., 2021) has emerged as a way to address image denoising. In this approach, a Vision Transformer (Dosovitskiy et al., 2020) is used to reconstruct missing input patches. The Masked Autoencoders (MAE) architecture (He et al., 2021), for example, efficiently reconstructs missing patches in pixel space and achieves strong performance on large labeled datasets. Other approaches, such as BEiT (Bao et al., 2021), predict a latent code obtained using a pretrained tokenizer. However, pixel-level pre-training has been shown to outperform BEiT in fine-tuning. SimMiM (Xie et al., 2021) explores simple reconstruction targets like color clusters but shows no significant advantages over pixel space reconstruction. Recently, Image-JEPA (I-JEPA) (Assran et al., 2023; LeCun, 2022) was proposed as a non-generative approach for self-supervised learning of semantic image representations. I-JEPA predicts the representations of various target blocks in an image from a single context block to guide it toward producing semantic representations. Our approach builds on this line of work and we propose to deal with location uncertainty using stochastic positional embeddings which was not explored before.

**Positional Embeddings in Transformers.** One of the core components of the Transformer architecture (Vaswani et al., 2017) is the Self-Attention block, which is a permutation invariant function, e.g, changing the order of the input tokens does not change the function output. Consequently, it is necessary to feed input tokens together with their positional embedding to describe their location. Absolute positional embeddings like fixed 2D sinusoidal features (Bello et al., 2019) or learned location features are the prevalent type of positional embeddings for the Vision Transformer (Dosovitskiy et al., 2020). Relative positional embeddings have recently gained popularity in NLP due to their ability to address the gap between the training and testing sequence length (Su et al., 2021; Chu et al., 2021; Press et al., 2021). For example, Press et al. (2021) proposed ALiBi to bias self-attention to assign higher confidence to neighboring locations, and SPE (Liutkus et al., 2021) proposed a stochastic approximation for relative positional embedding in linear transformers. Differently, we propose to use stochastic positional embeddings to tackle location uncertainties in MIM, and our approach can be easily applied on top of any existing deterministic variant.

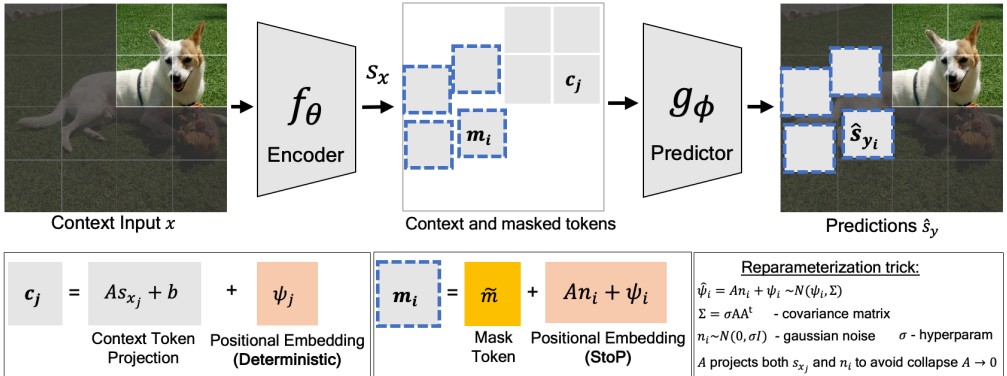

Figure 2: **Masked image modeling using stochastic positional embeddings (StoP).** The predictor $g_\phi$ predicts target tokens given masked tokens with stochastic positions $m_i$ and context tokens $c_j$ that were obtained via $f_\theta$. StoP is applied to masked tokens only, leading to features that are more robust to location uncertainties.

**Invariance-based methods.** These methods incorporate a loss that encourages similarity between augmented views of the the same image while avoiding a trivial solution. For example, contrastive learning prevents collapse by introducing negative examples (Hadsell et al., 2006; Dosovitskiy et al., 2014; Chen et al., 2020a; He et al., 2019; Chen et al., 2020b; Dwibedi et al., 2021). This can be achieved using a memory bank of previous instances (Wu et al., 2018; Oord et al., 2018; Tian et al., 2019; Misra & van der Maaten, 2020). However, there are also non-contrastive solutions that have been proposed. Of particular interest, a momentum encoder has been shown to prevent collapse even without negative pairs (Grill et al., 2020; Caron et al., 2021; Salakhutdinov & Hinton, 2007). Other methods include stopping the gradient to one branch (Chen & He, 2021) or applying regularization using batch statistics (Zbontar et al., 2021; Bardes et al., 2021; 2022; Ermolov et al., 2020; Hua et al., 2021). MoCo v3 Chen et al. (2021), then DiNO (Caron et al., 2021) extended these approaches for Vision Transformer, and iBOT (Zhou et al., 2021) proposed to add a MIM loss to DiNO. These approaches perform extremely well on ImageNet linear-probing, yet they rely on batch statistics, struggle under non-uniform distributions (Assran et al., 2022), and require hand-crafted image augmentations (Xiao et al.). Our approach is based on MIM, an alternative learning paradigm requiring less assumptions on batch statistics or handcrafted invariances.

## 3 MASKED IMAGE MODELING WITH STOP

We start by describing our stochastic positional embeddings (StoP) approach in Section 3.1, and then describe how to train a MIM with StoP in Section 3.2. A high-level schematic view of the model is included in Figure 2, and a pseudo-code implementation is included in Algorithm 1.

### 3.1 STOCHASTIC POSITIONAL EMBEDDING (STOP)

In Vision Transformers, the position of a patch $i$ is encoded via an embedding vector $\psi_i$. A common choice is to learn a vector embedding of the patch position or to use a fixed sine and cosine location features in different frequencies (Vaswani et al., 2017; Dosovitskiy et al., 2020). To reduce overfitting to location features, we wish to replace this deterministic mapping with a stochastic map. This involves a few crucial steps, including defining the distribution of the stochastic positions, parameterizing it appropriately, and implementing measures to prevent the model from reducing the impact of the noise to the point where it becomes negligible.

Given a position $i$, we denote by $\hat{\psi}_i$ the random variable providing the position embedding. We assume that $\hat{\psi}_i$ is distributed as Gaussian whose mean is the fixed embedding $\psi_i$, and covariance matrix $\Sigma \in \mathbb{R}^{d_P \times d_P}$:

$$\hat{\psi}_i \sim N(\psi_i, \Sigma) \tag{1}$$

Naturally, we want to learn an optimal $\Sigma$. However, this is challenging for two reasons. First, learning might result in the optimization process setting the values of $\Sigma$ to zero, leading to no

randomness. We refer to this case as a *degenerate determinism solution*. Second, the sampling process of $\hat{\psi}$ is non-differential, and therefore we cannot derive gradients to directly optimize it with SGD.

To solve these issues, we start by paramertizing $\Sigma$, then describe how to avoid degenerate determinism, and the reparametrization trick to derive a differential algorithm. We start by parameterizing $\Sigma$, and use a general formulation of a low-rank covariance matrix:

$$\Sigma = \sigma A A^T \tag{2}$$

Where $A \in \mathbb{R}^{d_p \times d_e}$ is a learned matrix and $\sigma \in \mathbb{R}^+$ is a positive scalar hyperparameter used to control the Noise to Signal Ratio (NSR). By learning matrix $A$, this formulation is flexibile enough, e.g, it is possible learning to assign small variance to low-res location features, while assigning higher variance to higher-frequency features, and also capturing correlations between location features.

**Reparametrization Trick.** Since $\hat{\psi}$ is sampled from a parameterized distribution, it is not immediately clear how to optimize over the learned parameters of the distribution $A$, because the sampling operation is non-differentiable in $A$. However, a standard trick in these cases is to reparameterize the distribution so that only sampling is from a fixed distribution that does not depend on $A$ (e.g., see Kingma & Welling (2013)). Specifically, we generate samples from $\hat{\psi}$ by first sampling a vector $n_i \in \mathbb{R}^{d_e}$ from a standard Gaussian distribution: $n_i \sim N(0, \sigma I)$. Then we set $\hat{\psi}$ to the following function:

$$\hat{\psi}_i = A n_i + \psi_i \tag{3}$$

The resulting distribution of $\hat{\psi}$ is equal to that in Equation 1, however, we can now differentiate directly through $A$.

**Avoiding a degenerate deterministic solution.** Without posing any constraints on $A$, it is easy for the model to scale down the noise by setting $A = 0$, thus $\hat{\psi} = \psi$, making the prediction problem deterministic again, and thereby easier. This would collapse the positional embedding back to the deterministic case, and we will lose the advantage of noisy spatial predictions. The main idea is to prevent this is to regularize $A$ and we describe this in more detail in Section 3.2 ("Avoiding a degenerate deterministic solution"), and analyze this further in Section 4.3.

## 3.2 MASKED IMAGE MODELING WITH STOP

Next, we proceed to describe in more detail how to apply StoP to Masked Image Modeling (MIM), see pseudo code impl. in Algorithm 1. The main idea in MIM is to predict target masked tokens based on contextual blocks from the same image. We introduce StoP in "Masked tokens in stochastic locations".

**Patchification.** Given an image, we apply the standard tokenization proposed by Dosovitskiy et al. (2020). Specifically, given an input image $I_x \in \mathbb{R}^{H \times W \times 3}$, it is first patchified into a sequence of non-overlapping image patches $\hat{p} = (\hat{p}_1, ..., \hat{p}_k)$ where $\hat{p}_i \in \mathbb{R}^{H' \times W' \times 3}$ and $K = \frac{HW}{H'W'}$ is the number of patches. Then, each patch is projected to $\mathbb{R}^{d_e}$ through a linear fully connected layer. Next, for every patch $\hat{p}_i$ the positional embedding features of the $i^{th}$ token are added to it, resulting in the patchified set $p = \{p_1, ...p_K\}$.

**Masking.** Let $x = \{p_i | i \in B_x\}$ be the set of context patches where $B_x$ denotes the set of context indices (e.g, the visible tokens in Figure 2). We denote by $B_y$ the indices of the target patches $y$. The context and target patches are chosen via random masking as in He et al. (2021) or by sampling target continuous blocks as in Assran et al. (2023).

---

**Algorithm 1** MIM w/ StoP pseudo-code.

1: **Input:** num iterations $K$, image dist $S$, hyperparam $\sigma$, positional embeddings $\psi$
2: **Params:** $A, b, \tilde{m}$, encoder $f_\theta$, predictor $g_\phi$
3: **for** $i = 1, 2, ..., K$ **do**
4:      $I_x \sim S$
5:      $p \leftarrow \text{patchify}(I_x)$
6:      $(x, B_x), (y, B_y) \leftarrow \text{mask}(p)$
7:      $s_x \leftarrow f_\theta(x)$
8:      # apply StoP
9:      $n \sim \mathcal{N}(0, \sigma I)$
10:     $m = An + \psi_{B_y} + \tilde{m}$
11:     $c = As_x + b + \psi_{B_x}$
12:     # predict targets
13:     $\hat{s}_y \leftarrow g_\phi(c, m)$
14:     $s_y \leftarrow \text{get\_target}(y)$
15:     loss $\leftarrow L(\hat{s}_y, s_y)$
16:     sgd\_step(loss; $\{\theta, \phi, A, b, \tilde{m}\}$)
17: **end for**

---

Algorithm 2: **Masked Image Modeling with StoP**. In practice, adding StoP to MIM requires only a minor implementation change, highlighted in light gray.

**Context Encoding.** First, the context tokens are processed via an encoder model $f_\theta$ to obtain deep representations: $s_x = f_\theta(x)$, Where $s_{x_i} \in \mathbb{R}^{d_e}$ is the $i^{th}$ context token representation.

**Masked tokens in stochastic locations.** We then define the set of context and masked tokens:

$$c_i = \psi_i + As_{x_i} + b \qquad\qquad m_j = \psi_j + An_j + \tilde{m}$$

Note that here the masked token $m_j$ has a stochastic position obtained via the reparametrization trick (see Eq. 3), while the context token $c_i$ has a deterministic position. $\tilde{m}$ is a learned bias used to signify a masked token, and it is shared across all masked token positions.

**Avoiding a degenerate deterministic solution.** Importantly, the matrix $A$ is used to linearly project every context token $s_{x_i}$ and every noise token $n_j$. The motivation for using A to project both the context features and noise can be understood by considering two extreme cases: When $A = 0$, there is complete certainty about the positional embeddings but all context is lost ($As_{x_i} = 0$). On the other hand, when $A$ is large the context information is preserved, but due to the large magnitude of $A$ the noise is amplified and camouflages the positional embedding features of the masked tokens: $An_j + \psi_j$. This dual role of matrix A forces the model to balance between location certainty and the influence of context features in predictions. It optimizes the trade-off for each feature, balancing their presence in predictions against the need for precise spatial locations. [1] This can also be viewed as a regularization of $A$ and we analyze this in Section 4.3.

**Prediction and Loss.** Then, we can apply a predictor function $g_\phi$ to predict the target features $\hat{s}_y = g_\phi(c, m)$. To supervise the prediction, the ground truth $s_y = \{s_{y_i}\}_{i \in B_y}$ is obtained either by using the raw RGB pixels or via a latent representation of the pixels. The loss $\frac{1}{|B_y|} \sum_{i \in B_y} L(s_{y_i}, \hat{s}_{y_i})$ is then applied to minimize the prediction error.

### 3.3 OPTIMAL PREDICTOR

Our approach relies on using stochastic positional embeddings. Here we provide further analysis of this prediction setting and show that the optimal prediction is indeed to perform spatial smoothing. Consider a random variable $X$ (corresponding to the context in our case. For simplicity assume $X$ is just the positional embedding of the context) that is used to predict a variable $Y$ (corresponding to the target in our case). But now instead of predicting from $X$, we use a noise variable $Z$ that is independent of both $X, Y$, and provide the predictor with only the noisy result $R = g(X, Z)$. Here $g$ is some mixing function (in our case $g(x, z) = x + z$). We next derive the optimal predictor $f(R)$ in this case. Formally we want to minimize:

$$E_{R,Y}[(f(R) - Y)^2] \qquad\qquad (4)$$

**Proposition 1.** *If $Z$ is a Gaussian with zero mean and unit variance, the optimal predictor that minimizes Equation 4 is:*

$$f(r) = \int_x E[Y|X = x] \frac{1}{\sqrt{2\pi}} e^{-0.5(x-r)^2} dx$$

Therefore, the optimal predictor amounts to a convolution of the clean expected values with a Gaussian. See Appendix A for a proof of this proposition.

## 4 EXPERIMENTS AND RESULTS

Next, we turn to discuss the main experiments presented in the paper. In Section 4.1, we describe the application of StoP to various downstream tasks including image recognition, dense prediction, and low-level vision tasks. In Section 4.2 we discuss the ablation study and design choices.

### 4.1 DOWNSTREAM TASKS

We conducted pre-training of StoP on top of I-JEPA, which is a state-of-the-art MIM model. We train on IN-1k for a period of 600 epochs using ViT-B/16 and ViT-L/16 architectures for the encoder

---

[1] Note that an implicit assumption here is that $\psi$ and $s_x$ have fixed magnitude. This is true for sine-cosine features and for $s_x$ which are layer normalized by the transformer last layer.

| Arch | Method | 1%, last layer | 100%, last layer | 100%, last 4 layers |
|------|--------|----------------|------------------|---------------------|
| ViT-B/16 | I-JEPA | 57.1 | 70.9 | 72.9 |
| | +StoP | 60.3 (+3.2%) | 72.6 (+1.7%) | 74.5 (+1.6%) |
| ViT-L/16 | I-JEPA | 64.2 | 76.1 | 77.5 |
| | +StoP | 65.1 (+0.9%) | 77.1 (+1.0%) | 78.5 (+1.0%) |
| ViT-H/14 | I-JEPA | 62.9 | 78.2 | 79.3 |
| | +StoP | 65.4 (+2.5%) | 79.0 (+0.8%) | 79.6 (+0.3%) |

Table 1: **Using StoP compared to deterministic sinusoidal positional embeddings on IN-1k**. StoP leads to consistent linear probing improvement in all settings. For example, when applying linear probing on trained ViT-H model with StoP, using only $1\%$ of the labeled data and using averaged pooled features from the last layer, StoP achieves $+2.5\%$ improvement. I-JEPA uses sinusoidal positional embeddings.

and predictor or for 300 epochs when using ViT-H/14. Subsequently, we proceeded to evaluate the model's performance on a variety of downstream tasks. We include the full implementation details, and provide additional results and comparisons to other approaches (e.g, invariance-based approaches) in Appendix B.

**Image recognition.** For image classification, we evaluated the StoP model linear probing performance on multiple datasets, including ImageNet (IN-1k) (Russakovsky et al., 2015), Places 205 (Zhou et al., 2014a), iNaturalist 2018 (Van Horn et al., 2018), and CIFAR 100 (Krizhevsky, 2009). These datasets vary in their size, their purpose, and the geographical environments from which the images were captured. For example, IN-1k contains over $1.2$ million images compared to CIFAR-100 which contains only $60,000$ images, and while IN-1k is focused on object recognition, Places is focused on scene recognition.

In Table 1, we present the linear probing image classification results conducted on IN-1k under different linear evaluation protocols using different amounts of data, and by aggregating features from different layers. E.g, "100%, last 4 layers" applies linear probing on the entire IN-1k data and the representation of each image is comprised of a concatenation of four feature vectors, each one summarizes information from its corresponding layer via average pooling. In Table 2 we compare linear probing results of common MIM methods on IN-1k, reporting past published performance. In Table 2 all perform linear probing over the output from the last layer.

StoP leads to consistent gains using all architectures. For example, $+2.5\%$ linear probing performance gains with ViT-H using $1\%$ of the labeled data and $1.6\%$ when using features from the last $4$ layer using ViT-B on the full IN-1k data. Furthermore, using StoP leads to improvements in downstream linear probing tasks (see Table 4). For example, StoP leads to $3.3\%$ improvement on iNAT using ViT-H and $1.3\%$ on counting. This confirms that the learned representations lead to improvements in a large variety of image recognition tasks. On full finetuning using $1\%$ of the labeled data, we observe similar performance improvements (see Table 5), e.g, $+2.3\%$ improvements on Top-1 accuracy using ViT-L model. We provide the full finetuning results in the Appendix.

| Method | Arch. | Epochs | Top-1 |
|--------|-------|--------|-------|
| data2vec | ViT-L/16 | 1600 | 77.3 |
| MAE | ViT-B/16 | 1600 | 68.0 |
| | ViT-L/16 | 1600 | 76.0 |
| | ViT-H/14 | 1600 | 76.6 |
| I-JEPA | ViT-B/16 | 600 | 70.9 |
| | ViT-L/16 | 600 | 76.1 |
| | ViT-H/14 | 300 | 78.2 |
| +StoP (ours) | ViT-B/16 | 600 | 72.6 |
| | ViT-L/16 | 600 | 77.1 |
| | ViT-H/14 | 300 | **79.0** |

Table 2: **Linear-evaluation on IN-1k**. Replacing sinusoidal positional embeddings with StoP in I-JEPA significantly improves linear probing results.

| Method | Arch. | J-Mean | F-Mean | J&F Mean |
|--------|-------|--------|--------|----------|
| MAE | ViT-B/16 | 49.4 | 52.6 | 50.9 |
| | ViT-L/16 | 52.5 | 54.3 | 53.4 |
| | ViT-H/14 | 54.0 | 57.0 | 55.5 |
| I-JEPA | ViT-B/16 | 56.1 | 56.2 | 56.1 |
| | ViT-L/16 | 56.1 | 55.7 | 55.9 |
| | ViT-H/14 | 58.5 | 60.9 | 59.7 |
| +StoP | ViT-B/16 | 56.6 | 57.3 | 57.0 |
| | ViT-L/16 | 58.1 | 58.7 | 58.4 |
| | ViT-H/14 | **58.9** | **61.2** | **60.1** |

Table 3: **Video objects semi-supervised segmentation.** MIM with StoP learn features in a finer level of granularity. Results reported on DAVIS 2017 dataset.

| Method | Arch. | CIFAR100 | Places205 | iNat18 | CLEVR/Count | CLEVR/Dist |
|--------|-------|----------|-----------|--------|-------------|------------|
| data2vec | ViT-L/16 | 81.6 | 54.6 | 28.1 | 85.3 | 71.3 |
| MAE | ViT-B/16 | 68.1 | 49.2 | 26.8 | 86.6 | 70.8 |
| | ViT-L/16 | 77.4 | 54.4 | 33.0 | **92.1** | 73.0 |
| | ViT-H/14 | 77.3 | 55.0 | 32.9 | 90.5 | 72.4 |
| I-JEPA | ViT-B/16 | 69.2 | 53.4 | 43.4 | 82.2 | 70.7 |
| | ViT-L/16 | 83.6 | 56.5 | 48.4 | 85.6 | 71.2 |
| | ViT-H/14 | 87.5 | 58.4 | 47.6 | 86.7 | 72.4 |
| +StoP | ViT-B/16 | 81.2 | 54.3 | 44.7 | 83.7 | 71.3 |
| | ViT-L/16 | 84.7 | 57.2 | 49.2 | 85.7 | 70.2 |
| | ViT-H/14 | **87.7** | **58.4** | **50.9** | 88.0 | **72.5** |

Table 4: **Linear-probe transfer for various downstream tasks**. Linear-evaluation on downstream image classification, object counting, and depth ordering tasks. Using StoP instead of sinusoidal deterministic positions leads to improvements on all tasks. E.g, +3.3% on iNAT18 and +1.3% on Counting.

**Counting and depth ordering.** We assess the downstream performance on tasks that require fine-grained objects representations like counting and depth ordering using the CLEVR (Johnson et al., 2017) dataset. Table 4 provides evidence that the representations learned by StoP significantly improve counting (+1.3%) and slightly improve depth ordering (+0.1%).

**Dense prediction.** To evaluate how well StoP performs on dense prediction tasks, e.g, tasks that require fine-grained spatial representations, we utilized the learned models for semi-supervised video object segmentation on the DAVIS 2017 (Pont-Tuset et al., 2017) dataset. We follow previous works (e.g Jabri et al. (2020); Caron et al. (2021)) and use the pretrained model to extract frames features and use patch-level affinities between frames to track the first segmentation mask. We include video semi-supervised video-object segmentation by tracking results in Table 3. We find that StoP significantly improves over I-JEPA with deterministic sinusoidal location features. For example, we observe +2.5% $J\&F$ improvement using ViT-L.

## 4.2 ABLATION STUDY

Our primary focus is to evaluate the effectiveness of stochastic positional embeddings (StoP). To demonstrate this, we evaluated various design options. For each setting, we implemented the encoder and predictor using ViT-B architecture and pretrained them for 300 epochs on IN-1k based on the I-JEPA (Assran et al., 2023) MIM model. We then assessed the linear probing performance on IN-1k using only 1% of the labels.

| Method | Epochs | Top-1 |
|--------|--------|-------|
| Sine Cosine | 600 | 69.4 |
| StoP (ours) | 600 | **71.7** |

Table 5: **Finetuning results over IN-1k with 1% labels.** Adding StoP to I-JEPA significantly improves finetuning using ViT-L/16 architecture.

**StoP compared to deterministic positional embeddings.**
The most common choices for positional embeddings for Vision Transformers are sine-cosine location features (also used in MAE, I-JEPA) and learned positional embedding. We evaluate the MIM downstream performance using each of these options and using StoP (see Table 6). Our results confirm that using StoP leads to significant (+3.2%) improvement compared to all counterparts.

**Learned vs. predefined covariance matrix.** To confirm that learning the covariance matrix $\Sigma = \sigma A A^T$ (and specifically $A$) is beneficial compared to using a predefined covariance matrix, we compare to stochastic positional embeddings with a predefined covariance matrix $\Sigma = \sigma I$, without any learning. We compare both options using different $\sigma$ hyperparameter values. Figure 3 indicates that it is advantageous to learn $\Sigma$ rather than use fixed parameters. Our findings show that setting the hyperparameter value to $\sigma = 0.25$ leads to an improvement of 3.5% points compared to deterministic positional embeddings ($\sigma = 0$).

**Application of StoP to different tokens.** We apply StoP to context and/or masked tokens. The results in Table 7 confirm our design choice, showing that StoP is most beneficial when applied solely to masked tokens positional embeddings and not to the context tokens.

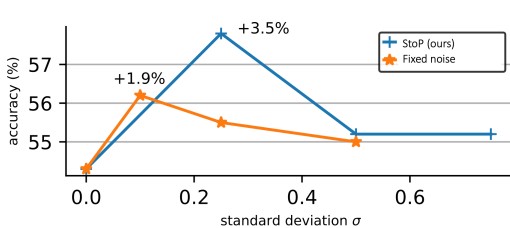
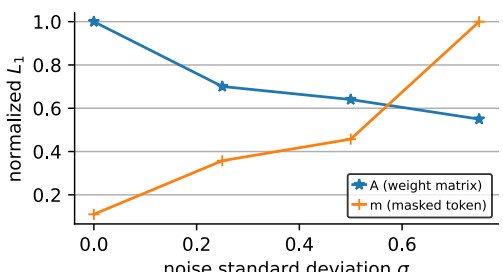

Figure 3: **Learned vs. predefined stochastic positions.** Using learned covariance matrix as in StoP, e.g, $\Sigma = \sigma A A^T$ leads to $+3.5\%$ improvement compared to smaller gains with a fixed covariance matrix $\Sigma = \sigma I$. Accuracy is based on linear probing using $1\%$ of the data from IN-1k.

Figure 4: **Increasing $\sigma$ induces regularization.** Changing the prior $\sigma$ (where $\Sigma = \sigma A A^T$) induces regularization over $A$ and increases the norm of the masked token, likely to preserve its information relative to the added noise.

| Method | Top-1 |
|---|---|
| Sine Cosine | 54.3 |
| Learned Pos. Embedding | 54.4 |
| Stochastic Positions (StoP) | **57.8** |

Table 6: **Different positional embeddings**. Linear probing on IN-1K using only $1\%$ of the labels. Stochastic Positions (StoP) outperforms other common deterministic variants by $3.3\%$.

| Method | Top-1 |
|---|---|
| No Noise (Sine Cosine) | 54.3 |
| Context tokens only | 55.1 |
| Masked + context tokens | 56.8 |
| Masked tokens only | **57.8** |

Table 7: **Applying noise to different tokens**. Applying learned noise to context and/or masked tokens positional embeddings (sine-cosine). Linear probing accuracy using $1\%$ of the data from IN-1k.

## 4.3 ANALYSIS

To explain how stochastic positional embeddings affect MIM, we analyze the learned model weights, visualize the stochastic positional embeddings, and visualize the reconstructed image features.

**StoP induces regularization.** The matrix $A$ is used to project both noise tokens and context embedding tokens, therefore, we hypothesize that StoP implicitly regularized $A$. To test this hypothesis we train models using StoP changing only the hyperparam $\sigma$ (see Figure 4). We find that increasing the value of $\sigma$ leads to a decrease in the norm of $A$, which can be viewed as regularization. On the other hand, increasing $\sigma$ leads to an increase in the norm of the masked token bias $\tilde{m}$. We speculate that the masked token bias increases in scale to prevent losing its information relative to the noise.

To further analyze this phenomenon, we train additional models while applying $l_1$ regularization on $A$ while keeping the positional embeddings of masked tokens deterministic. We find that simply regularizing the predictor projection layer leads to $1.5\%$ improvement over no-noise. However, applying StoP leads to higher performance gains ($+3.5\%$). Therefore, we conclude that StoP can only be partially explained by regularization.

**Stochastic positional embedding visualization.**

To visualize how StoP affects the similarity between different positions, we plot the similarity matrix between a stochastic position embedding query and the predefined sine-cosine deterministic positions (Figure 5). With StoP, we find that query locations are more similar to a wider range of neighboring locations. We build on this observation and train models to investigate if StoP performance could be achieved through predicting lower-scale features. We trained models to predict features in both the original scale and a downscaled version by a factor of 2, using bilinear resizing and max pooling for

| Method | Top-1 |
|---|---|
| Sine Cosine | 54.3 |
| x2 Low res (bilinear resize) | 52.1 |
| x2 Low res (max pooling) | 54.1 |
| Stochastic Positions (StoP) | **57.8** |

Table 8: **Low resolution prediction**. We evaluated the performance of StoP against models that predict features on lower scales via max pooling or bilinear resizing. Reporting linear evaluation results on IN-1K using $1\%$ of the labels.

downscaling. However, we found that predicting lower scale features does not improve performance (see Table 8).

**Prediction visualization.** We include heatmap visualization to visualize the similarity of a predicted token to all other tokens within the same image (see Figure 6). For a given image, mask, and a masked patch of interest, we apply cosine similarity between the predicted patch and all other

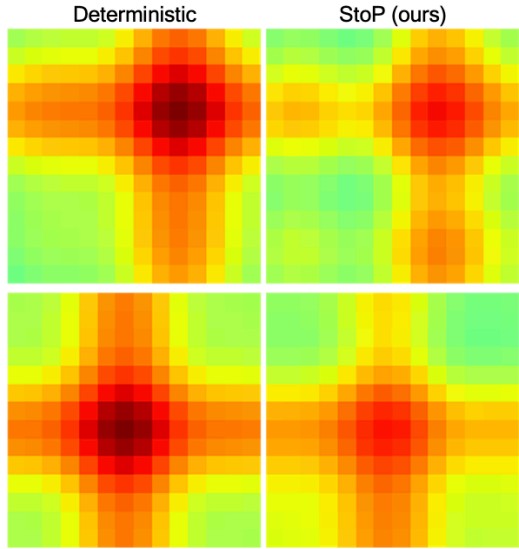

Figure 5: **Similarity matrices of deterministic and stochastic positional embedding (StoP) to a query position**. Each row represents the similarity given a different query position. StoP leads to spatially smooth similarity matrix, thereby making it hard to distinguish the exact location of a given patch.

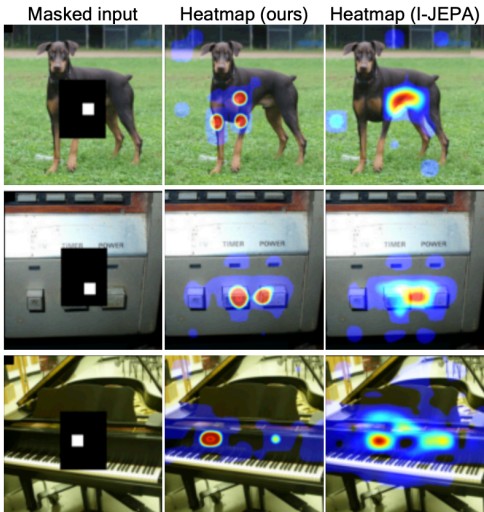

Figure 6: **Feature visualization**. We plot the similarity between the predicted features of a given patch (marked in white within the masked black area) and the other tokens in the same image. Using StoP produces features that are more semantic than location based. Patches predicted by I-JEPA tend to have strong correlation with the target location features.

token representations within the same image, followed by a softmax. For I-JEPA with sine-cosine positional embeddings, the visualization indicates that adjacent tokens tend to share similar features, implying a correlation between the features and spatial location. In contrast, StoP produces predictions correlated with non-neighboring small areas. We speculate that using StoP leads to learning features that are more semantic and prevents overfitting to location features.

## 5 LIMITATIONS

We applied StoP to I-JEPA which performs image reconstruction in the feature space. However, our attempts to apply StoP to MIM that use pixel based reconstruction, mainly MAE, were not successful. We speculate that adding StoP to MAE might make pixel reconstruction too difficult to achieve. Additionally, StoP tackles location uncertainty but not appearance uncertainty, which we believe is implicitly modeled by reconstructing tokens in feature space. Also, when modeling stochastic positions it may might be possible to condition the noise on the input image, namely the context tokens. We leave this extension for future work. Lastly, while combining StoP with MIM shows significant improvements, invariance-based approaches still perform slightly better (e.g, iBOT) than MIM approaches.

## 6 CONCLUSION

In this work, we proposed to use stochastic positional embedding (StoP) to tackles location uncertainties in the MIM task. By conditioning on stochastic masked tokens positions, our model learns features that are more robust to location uncertainties. The effectiveness of this approach is demonstrated on various datasets and downstream tasks, outperforming existing MIM methods and highlighting its potential for self-supervised learning. Based on our experiments and visualizations, by modeling location uncertainties with StoP, models suffer less from overfitting to location features.

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
