APPENDIX

## A OPTIMAL PREDICTOR

Consider a random variable $X$ (corresponding to the context in our case. For simplicity assume $X$ is just the positional embedding of the context) that is used to predict a variable $Y$ (corresponding to the target in our case). But now instead of predicting from $X$, we use a noise variable $Z$ that is independent of both $X, Y$, and provide the predictor with only the noisy result $R = g(X, Z)$. Here $g$ is some mixing function (in our case $g(x, z) = x + z$). We next derive the optimal predictor $f(R)$ in this case. Formally we want to minimize:

$$E_{R,Y}[(f(R) - Y)^2] \tag{4}$$

A classic result in estimation is that this is optimized by the conditional expectation $f(r) = E[Y|R = r]$.

We simplify this as follows:

$$
\begin{aligned}
E[Y|R = r] &= \sum_{x,y} y p(Y = y, X = x | R = r) \\
&= \sum_{x,y} y p(y|X = x) p(X = x | R = r) \\
&= \sum_{x} E[Y|X = x] p(X = x | R = r)
\end{aligned}
$$

where in the second line we used the fact that:

$$p(y, x|r) = p(y|x, r)p(x|r) = p(y|x)p(x|r) \tag{5}$$

To further illustrate, consider the case where $z$ is Gaussian with zero mean and unit variance. Then $p(x|r)$ is also Gaussian with expectation $r$, and the expression above amounts to convolution of the clean expected values with a Gaussian:

$$E[Y|R = r] = \int_x E[Y|X = x] \frac{1}{\sqrt{2\pi}} e^{-0.5(x-r)^2} dx \tag{6}$$

## B EXPERIMENTS AND RESULTS

We include the full implementation details, pretraining configs and evaluation protocols for the Ablations (see Appendix B.1), Downstream Tasks (Appendix B.2), as well as full results and comparisons to invariance-based methods.

### B.1 ABLATIONS

Here we pretrain all models for 300 epochs using 4 V100 nodes, on a total batch size of 2048. In all the ablation study experiments, we follow the exact recipe of Assran et al. (2023). We include the full config in Table 9 for completeness.

To evaluate the pretrained models, we use linear probing evaluation using 1% of IN-1k (Russakovsky et al., 2015). To obtain the features of an image, we apply the target encoder over the image to obtain a sequence of tokens corresponding to the image. We then average the tokens to obtain a single representative vector. The linear classifier is trained over this representation, maintaining the rest of the target encoder layers fixed.

### B.2 DOWNSTREAM TASKS

Here we pretrain I-JEPA with StoP for 600 epochs using 4 V100 nodes, on a total batch size of 2048 using ViT-B (see config in Table 10) and ViT-L (see config in Table 11). For ViT-H we use float16 and train for 300 epochs and follow the config in Table 12. We follow similar configs

compared to Assran et al. (2023) except we usually use a lower learning rate. Intuitively, since StoP is stochastic it is more sensitive to high learning rates.

For evaluation on downstream tasks, we use the features learned by the target-encoder and follow the protocol of VISSL Goyal et al. (2021) that was utilized by I-JEPA Assran et al. (2023). Specifically, we report the best linear evaluation number among the average-pooled patch representation of the last layer and the concatenation of the last 4 layers of the average-pooled patch representations. We report full results including comparisons to invariance-based methods for IN-1k linear evaluation Table 14, 1% IN-1k finetuning results in Table 16, and other downstream tasks in Table 13.

For baselines that use Vision Transformers Dosovitskiy et al. (2020) with a `[cls]` token (e.g, iBOT Zhou et al. (2021), DINO Caron et al. (2021) or MAE He et al. (2021)), we use the default configurations of VISSL Goyal et al. (2021) to evaluate the publicly available checkpoints on iNaturalist18 Van Horn et al. (2018), CIFAR100 Krizhevsky et al. (2009), Clevr/Count Johnson et al. (2017); Zhai et al. (2019), Clevr/Dist Johnson et al. (2017); Zhai et al. (2019), and Places205 Zhou et al. (2014b). Following the evaluation protocol of VISSL Goyal et al. (2021), we freeze the encoder and return the best number among the `[cls]` token representation of the last layer and the concatenation of the last 4 layers of the `[cls]` token.

For semi-supervised video object segmentation, we propagate the first labeled frame in a video using the similarity between adjacent frames features. To label the video using the frozen features, we follow the code and hyperparams of Caron et al. (2021). To evaluate the segmented videos, we use the evaluation code of DAVIS 2017 (Pont-Tuset et al., 2017) and include full results in Table 15.

| config | value |
|---|---|
| optimizer | AdamW |
| epochs | 300 |
| learning rate | $1e^{-3}$ |
| weight decay | $(0.04, 0.4)$ |
| batch size | 2048 |
| learning rate schedule | cosine decay |
| warmup epochs | 15 |
| encoder arch. | ViT-B |
| predicted targets | 4 |
| predictor depth | 6 |
| predictor attention heads | 12 |
| predictor embedding dim. | 384 |
| $\sigma$ (noise hyperparam) | 0.25 |

Table 9: **Pretraining setting for ablations**. Using ViT-B encoder, trained for 300 epochs, config strictly follows Assran et al. (2023).

| config | value |
|---|---|
| optimizer | AdamW |
| epochs | 600 |
| learning rate | $8e^{-4}$ |
| weight decay | $(0.04, 0.4)$ |
| batch size | 2048 |
| learning rate schedule | cosine decay |
| warmup epochs | 15 |
| encoder arch. | ViT-B |
| predicted targets | 4 |
| predictor depth | 6 |
| predictor attention heads | 12 |
| predictor embedding dim. | 384 |
| $\sigma$ (noise hyperparam) | 0.25 |

Table 10: **Pretraining setting for downstream tasks (ViT-B)**. All models trained for 600 epochs.

| config | value |
|---|---|
| optimizer | AdamW |
| epochs | 600 |
| learning rate | $8e^{-4}$ |
| weight decay | $(0.04, 0.4)$ |
| batch size | 2048 |
| learning rate schedule | cosine decay |
| warmup epochs | 15 |
| encoder arch. | ViT-L |
| predicted targets | 4 |
| predictor depth | 12 |
| predictor attention heads | 16 |
| predictor embedding dim. | 384 |
| $\sigma$ (noise hyperparam) | 0.25 |

Table 11: **Pretraining setting for downstream tasks (ViT-L)**. All models trained for 600 epochs.

| config | value |
|---|---|
| optimizer | AdamW |
| epochs | 600 |
| learning rate | $1e^{-3}$ |
| weight decay | $(0.04, 0.4)$ |
| batch size | 2048 |
| learning rate schedule | cosine decay |
| warmup epochs | 40 |
| encoder arch. | ViT-H |
| predicted targets | 4 |
| predictor depth | 12 |
| predictor attention heads | 16 |
| predictor embedding dim. | 384 |
| $\sigma$ (noise hyperparam) | 0.2 |

Table 12: **Pretraining setting for downstream tasks (ViT-H)**. Trained for 300 epochs.

| Method | Arch. | CIFAR100 | Places205 | iNat18 | CLEVR/Count | CLEVR/Dist |
|---|---|---|---|---|---|---|
| *Invariance-based methods (use extra image augmentations)* | | | | | | |
| DINO | ViT-B/16 | 84.8 | 55.2 | 50.1 | 83.2 | 53.4 |
| iBOT | ViT-B/16 | 85.5 | 56.7 | 50.0 | 62.1 | 64.6 |
| | ViT-L/16 | 88.3 | 60.4 | 57.3 | 85.7 | 62.8 |
| *Masked Image Modeling Methods* | | | | | | |
| data2vec | ViT-L/16 | 81.6 | 54.6 | 28.1 | 85.3 | 71.3 |
| MAE | ViT-B/16 | 68.1 | 49.2 | 26.8 | 86.6 | 70.8 |
| | ViT-L/16 | 77.4 | 54.4 | 33.0 | **92.1** | 73.0 |
| | ViT-H/14 | 77.3 | 55.0 | 32.9 | 90.5 | 72.4 |
| I-JEPA | ViT-B/16 | 69.2 | 53.4 | 43.4 | 82.2 | 70.7 |
| | ViT-L/16 | 83.6 | 56.5 | 48.4 | 85.6 | 71.2 |
| | ViT-H/14 | 87.5 | 58.4 | 47.6 | 86.7 | 72.4 |
| +StoP | ViT-B/16 | 81.2 | 54.3 | 44.7 | 83.7 | 71.3 |
| | ViT-L/16 | 84.7 | 57.2 | 49.2 | 85.7 | 70.2 |
| | ViT-H/14 | **87.7** | **58.4** | **50.9** | 88.0 | **72.5** |

Table 13: **Linear-probe transfer for various downstream tasks**. Linear-evaluation on downstream image classification, object counting, and tracking tasks. StoP significantly outperforms previous MIM methods that don't utilize image augmentations like I-JEPA and MAE, and decreases the gap with the best invariance-based methods that utilize data augmentations during pretraining.

| Method | Arch. | Epochs | Top-1 |
|---|---|---|---|
| *Invariance-based methods (use extra image augmentations)* | | | |
| SimCLR v2 | RN152 (2×) | 800 | 79.1 |
| BYOL | RN200 (2×) | 800 | 79.6 |
| DINO | ViT-B/16 | 400 | 78.1 |
| | ViT-B/8 | 300 | 80.1 |
| MoCo v3 | ViT-B/16 | 300 | 76.7 |
| | ViT-BN-L/7 | 300 | 81.0 |
| MSN | ViT-L/7 | 200 | 80.7 |
| iBOT | ViT-B/16 | 250 | 79.8 |
| | ViT-L/16 | 250 | 81.0 |
| *Masked Image Modeling methods* | | | |
| data2vec | ViT-L/16 | 1600 | 77.3 |
| MAE | ViT-B/16 | 1600 | 68.0 |
| | ViT-L/16 | 1600 | 76.0 |
| | ViT-H/14 | 1600 | 77.2 |
| I-JEPA | ViT-B/16 | 600 | 72.9 |
| | ViT-L/16 | 600 | 77.5 |
| | ViT-H/14 | 300 | 79.3 |
| +StoP (ours) | ViT-B/16 | 600 | 74.5 |
| | ViT-L/16 | 600 | 78.5 |
| | ViT-H/14 | 300 | **79.6** |

Table 14: **Linear-evaluation on IN-1k**. Performance of invariance based and MIM approaches.

| Method | Arch. | J-Mean | F-Mean | J&F Mean |
|---|---|---|---|---|
| *Invariance-based methods (use extra image augmentations)* | | | | |
| DINO | ViT-B/16 | 60.7 | 63.9 | 62.3 |
| iBOT | ViT-B/16 | 60.9 | 63.3 | 62.1 |
| | ViT-L/16 | 61.7 | 63.9 | 62.8 |
| *Masked Image Modeling Methods* | | | | |
| MAE | ViT-B/16 | 49.4 | 52.6 | 50.9 |
| | ViT-L/16 | 52.5 | 54.3 | 53.4 |
| | ViT-H/14 | 54.0 | 57.0 | 55.5 |
| I-JEPA | ViT-B/16 | 56.1 | 56.2 | 56.1 |
| | ViT-L/16 | 56.1 | 55.7 | 55.9 |
| | ViT-H/14 | 58.5 | 60.9 | 59.7 |
| +StoP | ViT-B/16 | 56.6 | 57.3 | 57.0 |
| | ViT-L/16 | 58.1 | 58.7 | 58.4 |
| | ViT-H/14 | **58.9** | **61.2** | **60.1** |

Table 15: **Video objects semi-supervised segmentation.** MIM and Invarianced-based methods. Results reported on DAVIS 2017 dataset.

| Method | Arch. | Epochs | Top-1 |
|---|---|---|---|
| *Invariance-based methods (use extra image augmentations)* | | | |
| DINO | ViT-B/8 | 300 | 70.0 |
| iBOT | ViT-B/16 | 400 | 69.7 |
| *Masked Image Modeling methods* | | | |
| MAE | ViT-L/16 | 1600 | 67.0 |
| I-JEPA | ViT-L/16 | 600 | 69.4 |
| +StoP (ours) | ViT-L/16 | 600 | **71.7** |

Table 16: **Finetuning results over ImageNet with 1% labels.** Comparison of MIM and invariance-based methods.