# OpenReview forum: "Predicting masked tokens in stochastic locations improves masked image modeling"
_ICLR.cc/2024/Conference — Submitted to ICLR 2024_

### Official Review · Reviewer_gTwQ · 2023-10-20

**Soundness:** 2 fair
**Presentation:** 2 fair
**Contribution:** 2 fair
**Rating:** 5
**Confidence:** 3

**Summary:**

This paper proposes stochastic positional embeddings (StoP) to improve masked image modeling (MIM), which incorporates location uncertainty by conditioning the model on stochastic masked token positions drawn from Gaussian distribution. Experimental results demonstrate that using StoP reduces overfitting to location features and guides the model toward learning features that are more robust to location uncertainty, which also leads to better performance on a variety of downstream tasks.

**Strengths:**

- The idea of stochastic positional embedding proposed here is novel to me
- Experiments are sufficient to support the proposed method, showing that the proposed method can achieve significant improvements on various downstream tasks

**Weaknesses:**

Several parts of the proposed method are not properly introduced and may cause some confusions, details can be found in Questions part

**Questions:**

- I am a bit confused on step 11 in Algorithm 1. As in Figure 2, the context and masked representations are computed by adding their tokens and positional embeddings together. Then for step 11, I suppose $\psi_{B_x}$ should refer to the positional embedding, and $A s_x+b$ should refer to context token? Why do we need an additional linear transformation on $s_x$? Some explanations may be needed for this part.
- Based on the above concern, I am also confused by later explanations in section 3.2 and 4.3, The authors seem to let $s_{x_i}$ (resp. $n_j$) as context (resp. masked) tokens, and $b$ (resp. $\tilde{m}$) corresponds to the bias for context (resp. masked) tokens. However, I suppose $n_j$ should simply be used to compute stochastic positional embedding as in (2), and $s_{x_i}$ is computed from encoder $f_\theta$ to encode context information. How can they have the same role?
- Moreover, with the above correspondence, we should have $A s_x+b$ (resp. $An+\tilde{m}$) as context (resp.) tokens, then the positional embedding is simply $\psi_{B_x}$ (resp. $\psi_{B_y}$), and where is the stochasticity? I suppose there might be some misunderstanding.
- I would also like to see more discussions on the connection between StoP and vanilla MIM. I suppose we can replace step 10 with $\tilde{m} + \psi_{B_y}$, and step 11 with $s_x+\psi_{B_x}$ to reduce to vanilla MIM, is it correct? Such discussions may make it easier to understand the proposed method.
- While the authors have mentioned the necessity of regularization on A, the regularization with context token is a bit confusing. I note that the authors have conducted additional experiments in section 4.3 that uses L1 regularization on A. Nevertheless, L1 regularization should aim to obtain a sparse matrix A, which seems to contradict with the original aim to avoid zero A. The authors may consider using some other regularization (and also remove A in computing context tokens) and see how such modification works compared to Algorithm 1.

Minor: the authors may also need to pay more attention on notations and typos. An example is on the top of page 5 “Context Encoding”, “Where” is wrongly capitalized (in fact the capitalization is used very arbitrarily and may require a careful proof-reading). Also, the notation through this paper is not consistent, especially for representations $c$ and $m$. Some revisions may be needed as well.

---

> ### Author Response · Authors · 2023-11-13
> **Response to Reviewer gTwQ**
>
> Thank you for the thoughtful consideration of the paper and constructive feedback. We’ve incorporated your feedback and uploaded a new revision of the paper.
>
> **Q: “As in Figure 2, the context and masked representations are computed by adding their tokens and positional embeddings together”**
>
> Thank you for pointing this out, there is a mistake in Figure 2 and we apologize for the confusion. To clarify, the masked and context tokens are computed as follows (as in Algorithm 11):
>
> 11: $n \sim \mathcal{N}(0, \sigma I)$
>
> 12: $m = An + \psi_{B_y} + \tilde{m}$
>
> 13: $c = As_x + b + \psi_{B_x}$
>
> We uploaded a new paper revision and fixed Figure 2.
>
> **Q: “For step 11, I suppose $\psi_{B_x}$  should refer to the positional embedding, and
> $As_x + b$ should refer to context token?**
>
> You are right.
>
> **Q: Why do we need an additional linear transformation on $s_x$? Some explanations may be needed for this part.**
>
> The mapping projects $s_x$ from the output dimension of the encoder to the input dimension of the predictor (this is standard for other approaches like MAE and I-JEPA).
>
> **Q: The authors seem to let $s_{x_i}$ (resp. $n_j$) as context (resp. masked) tokens, and
> $b$ (resp. $\tilde{m}$) corresponds to the bias for context (resp. masked) tokens. However, I suppose $n_j$ should simply be used to compute stochastic positional embedding as in (2), and $s_{x_i}$ is computed from encoder $f_{\theta}$ to encode context information. How can they have the same role?”**
>
> We assume that you ask why both the context $s_{x_i}$ and noise $n_j$ linearly projected by matrix $A$. In the original MIM formulation there is no stochastic positions and the masked tokens and context tokens are computed as follows:
>
> 12: $m = \psi_{B_y} + \tilde{m}$
>
> 13: $c = Bs_x + b + \psi_{B_x}$
>
> When applying StoP with the reparameterization trick (Eq.3, revised manuscript), the noise is now linearly projected with a matrix $A$ (defined in Eq. 1) and summed with the positional embeddings. Therefore, both the sampled noise and the context tokens are now linearly projected:
>
> 11: $n \sim \mathcal{N}(0, \sigma I)$
>
> 12: $m = An + \psi_{B_y} + \tilde{m}$
>
> 13: $c = Bs_x + b + \psi_{B_x}$
>
> Note that here the context tokens and noise use different projections $A$ and $B$. However, we find that the weights of A are quickly scaled down during training, setting $An=0$ which overcomes the noise during training, resorting to the original MIM introduced before, and the same empirical downstream accuracy. We discussed this in Section 3.1 and 3.2 (see “Avoiding a degenerate determinism solution” and “Masked tokens in stochastic locations”.)
>
> To avoid this, we use the same matrix $A$ to also project the context tokens $s_x$ (line 13), instead of using a different projection matrix $B$:
>
> 11: $n \sim \mathcal{N}(0, \sigma I)$
>
> 12: $m = An + \psi_{B_y} + \tilde{m}$
>
> 13: $c = As_x + b + \psi_{B_x}$
>
> The motivation for using $A$ to project both the context features and noise can be understood by considering two extreme cases. When $A=0$, there is complete certainty about the positional embeddings but all context is lost ($As_{x}=0$). On the other hand, when $A$ is large the context information is preserved, but due to the large magnitude of $A$ the noise is amplified and camouflages the positional embedding features of the masked tokens: $An + \psi_{B_y}$. This dual role of matrix A forces the model to balance between location certainty and the influence of context features in predictions. It optimizes the trade-off for each feature, balancing their presence in predictions against the need for precise spatial locations.
>
> We discussed this in Section 3.2 in the original submission but following the comments we revised the manuscript to make this more clear.
>
> **Q: I would also like to see more discussions on the connection between StoP and vanilla MIM.**
>
> The above answer should clarify this comment as well. We follow your advice and clarify this in Algorithm 1 caption and highlight the differences in the Algorithm (see revised manuscript).

---

> > ### Author Response · Authors · 2023-11-13
> > **Response to Reviewer gTwQ (cont)**
> >
> > **Q: While the authors have mentioned the necessity of regularization on A, the regularization with context token is a bit confusing. I note that the authors have conducted additional experiments in section 4.3 that uses L1 regularization on A. Nevertheless, L1 regularization should aim to obtain a sparse matrix A, which seems to contradict with the original aim to avoid zero A. The authors may consider using some other regularization (and also remove A in computing context tokens) and see how such modification works compared to Algorithm 1.**
> >
> > To clarify, in the regularization experiments we followed a similar setting to what you suggested (no stochasticity). We used the basic MIM recipe plus regualrization over the projection matrix that projects the context tokens from the encoder output dimension to the predictor input dimension:
> >
> > $m = \psi_{B_y} + \tilde{m}$
> >
> > $c = As_x + b + \psi_{B_x}$
> >
> > Since there is no stochasticity, we do not need to worry about avoiding A going to zero to scale the noise down ($An=0$). There is still the tradeoff between the MIM reconstruction loss and regularization loss (but this is always the case with regularization).

---

> > > ### Author Response · Authors · 2023-11-19
> > > **Followup**
> > >
> > > Dear reviewer, towards the end of the discussion phase, we trust that our response has successfully addressed your inquiries. We look forward to receiving your feedback regarding whether our reply sufficiently resolves any concerns you may have, or if further clarification is needed.

---

> > > > ### Comment · Reviewer_gTwQ · 2023-11-21
> > > > **Acknowledging the responses**
> > > >
> > > > I suppose some of my previous concerns are successfully resolved, mainly on the connection between StoP and vanilla MIM. Nevertheless, I still have some confusions regarding some details of StoP:
> > > > - The use of using the same matrix $A$ for both projection and covariance still sounds strange to me. I understand that currently StoP effectively prevents $A=0$ as it will lead to $As_x=0$ (no context). Nevertheless, I suppose there might be some other implementations, a straight-forward idea is to use an additional matrix $B$ and computes $m = An+\psi_{B_y}+\tilde{m}, c = BAs_x+b+\psi_{B_x}$. In such case, $A=0$ also leads to $BAs_x=0$ (no context). I wonder if the authors can provide some discussions on that.
> > > > - I am now a bit confused on the experiments on regularization. I suppose you are trying to prove that the improvements of StoP do not solely come from regularizing $A$ (which is used as the projection matrix for context token)? However, I am not sure if the matrix $A$ in StoP is really regularized towards a sparse matrix. Given that you observed that the norm of $A$ decreases with increasing $\sigma$, I suppose you should try $\ell_2$ regularization (which regularizes the norm of matrix $A$) instead of $\ell_1$, and see if that can lead to much improvement.

---

> ### Author Response · Authors · 2023-11-21
>
> Dear reviewer, thank you for the reply and we are happy that some of your previous concerns are resolved.
>
> **Q: I suppose there might be some other implementations, a straight-forward idea is to use an additional matrix B**
>
> Thank you for this suggestion. The idea to use the matrix B would cancel the noise and lead to a deterministic solution (i.e., removing our novel noise component), and thus this is undesirable. For example:
> If  $A = \epsilon I$, then  the noise is scaled down via $An$ and the positional embedding $\psi_{B_y}$ is unaffected. B can then be set to be $B = A^{-1} = \frac{1}{\epsilon} I$, and this will preserve the context tokens information.
>
> There might be other ways to regularize A that can be explored, for example, by incorporating additional (multiple) loss terms that ensure A has a large enough norm, and that it is full rank. However, our solution is simpler as it doesn't require additional losses and hyperparam tuning.
>
> **Q: I am now a bit confused on the experiments on regularization. I suppose you are trying to prove that the improvements of StoP do not solely come from regularizing $A$? (which is used as the projection matrix for context token)? However, I am not sure if the matrix $A$ in StoP is really regularized towards a sparse matrix. Given that you observed that the norm of $A$ decreases with increasing $\sigma$, I suppose you should  try  ℓ2 regularization (which regularizes the norm of matrix $A$) instead of ℓ1, and see if that can lead to much improvement.**
>
> Indeed, we wanted to show that the improvements of StoP are not just due to reducing the norm of A. Clearly, there are several notions of norm, and these can be explored. We focused on $L_1$ because this is a standard approach to regularizing the rank of $A$ for the diagonal case. Furthermore, it is well known that optimization with SGD implicitly regularizes $L_2$ norm (e.g., see https://arxiv.org/abs/1906.05890), so we wanted to test a norm that is not implicitly regularized.
>
> We note that our $L_1$ regularization experiments also resulted in low $L_2$ (the higher the $L_1$ regularization loss coefficient , the lower the $L_2$ norm, see table below).
>
> | $L_1$ loss Coeff      | $L_2$ norm |
> | ----------- | ----------- |
> | 1.0     | 0.00002       |
> | 0.1  | 0.00007        |
> | 0.01 | 0.00010        |
> | 0.001 | 0.00020        |

---

> ### Author Response · Authors · 2023-11-22
>
> > The authors may consider using some other regularization (and also remove A in computing context tokens)
>
> > I suppose you should try ℓ2 regularization (which regularizes the norm of matrix $A$) instead of ℓ1, and see if that can lead to much improvement.
>
> Dear reviewer, we follow up on your suggestion and include additional experiments applying $L_2$ regularization on $A$.
>
> Specifically, we trained ViT-B/16 baseline models using deterministic sine-cosine positional embeddings for 150 epochs while adding $L_2$ regularization loss weighted by $\alpha \in${$1.0, 0.1, 0.01, 0.001$}. We then applied the ImageNet linear probing protocol, then report the results below.
>
> These results indicate that StoP cannot be merely replaced by $L_2$ regularization over $A$.
> Please let us know if there are any other concerns, and we are open to hear more feedback or provide further clarification if needed.
>
>
> | Model      | Top-1 Acc |
> | ----------- | ----------- |
> | Baseline, $\alpha=0.001$      | 61.7       |
> | Baseline, $\alpha=0.01$      | 62.7       |
> | Baseline, $\alpha=0.1$      |    61.9   |
> | Baseline, $\alpha=1.0$      | 59.8       |
> | StoP      | 64.8 (+2.1)       |

---

> > ### Author Response · Authors · 2023-11-23
> >
> > Dear reviewer, we would greatly appreciate it if you could review our new response. We believe that we have effectively addressed all of your previous concerns. We actively stand by for the last few hours of the discussion phase.

---

### Official Review · Reviewer_92Mr · 2023-10-30

**Soundness:** 3 good
**Presentation:** 3 good
**Contribution:** 3 good
**Rating:** 8
**Confidence:** 4

**Summary:**

The paper proposes modeling a distribution over positional embeddings instead of learning/using deterministic ones which is compatible with any Masked Image Modeling (MIM) framework.

**Strengths:**

Authors propose smart modeling design choice to avoid collapsing model to just learn deterministic embeddings. Experimental evaluation shows consistent improvements compared to deterministic MIM (i.e. I-JEPA) for models of different sizes. Also, ablation study is great, authors ablate and deeply study different aspects of the model.

**Weaknesses:**

Honestly, I don't see any obvious weaknesses of the work.

**Questions:**

To strengthen the evaluation, it would be nice to see linear probes/finetuning results on the larger set of downstream datasets. Also, it could be nice to have a model pretrained on a larger dataset rather than Imagenet-1000 as it could lead to stronger model and will enable better transfer to downstream problems which is important to have such representations for the community.

---

> ### Author Response · Authors · 2023-11-13
> **Response to Reviewer 92Mr**
>
> Thank you for the thoughtful consideration of the paper and very positive feedback.
>
> **Q: To strengthen the evaluation, it would be nice to see linear probes/finetuning results on the larger set of downstream datasets.**
>
> Thank you for the comment. We’ve evaluated StoP on 5 different datasets (ImageNet, iNat, Places, DAVIS 2017, CLEVR).  Following your comment, we will run additional evaluations on CUB-200, Flowers-102 and IN-100 and will include it in the final manuscript.
>
> **Q: it could be nice to have a model pretrained on a larger dataset rather than Imagenet-1000 as it could lead to stronger model and will enable better transfer to downstream problems which is important to have such representations for the community.**
>
> Thank you for the suggestion. We think that running large scale experiments (e.g, on LAION 5B) with StoP is exciting. Since this might require non trivial engineering efforts and amounts of resources, we leave this for future work.

---

> > ### Comment · Reviewer_92Mr · 2023-11-16
> >
> > I would like to thank the authors for the clarifications and will maintain my initial assessment of the paper.

---

> > > ### Author Response · Authors · 2023-11-20
> > >
> > > Thank you very much; we truly value your support in accepting the paper.

---

### Official Review · Reviewer_Hczn · 2023-10-31

**Soundness:** 2 fair
**Presentation:** 3 good
**Contribution:** 2 fair
**Rating:** 5
**Confidence:** 4

**Summary:**

The paper proposes the Stochastic Positionalem beddings (StoP) to MIM in order to perturb the location information of images as a way of regularization. This avoids overfitting the model. The paper motivates and derives the empirical training loss of such perturbation that allows for end to end training by borrowing the well known reparametrization trick. Empirical evidence shows that the proposed method improves the existing SOTA method by evident margin.

**Strengths:**

The paper has several strengths including:

S1. It introduces Stochastic Positional Embeddings (StoP) for the purpose of adding perturbations to the location information of images within the MIM framework, thus serving as a means of regularization. This measure intuitively can prevent the model from overfitting.

S2. By employing a reparametrization trick, the paper trivially both justifies and develops the empirical training loss associated with this form of perturbation, enabling end-to-end training.

S3. Empirical results highlight that this proposed technique significantly enhances the state-of-the-art method, demonstrating a noticeable improvement.

**Weaknesses:**

However, there are also several concerning points that needs to be addressed:

W1: It is unclear to me why it is necessary to learn optimal $\Sigma$ via additional parameterization. What is the benefits of introducing additional degree of freedom here to learn Sigma? What if we fix Sigma without learning? Isn't it a simpler way to avoid degeneracy of matrix A?  Please explain the motivation.

W2: I understand that adding stochastic perturbation to position of the images makes sense in regularizing the model. However, why the same spectral decomposition is applied to features s_x (by multiplying with A)? This step also lacks motivation and seems to be heuristic, please clarify on this point,

W3: What exactly architecture did the paper use to parameterize the matrix $\Sigma$ ? An architecture flow illustration will help better illustrate this mechanism. Currently, I am not sure how the back-propagation of $\Sigma$ flows back to the network  (figure 1 does not have this part ) and how it affects the SSL learning with a positive gain.

W4: I am not sure of the significance of proposition 1. I do not see why using this optimal predictor can help achieve better generalization ability of the SSL pretraining on downstream tasks.

**Questions:**

Please see above for the in total 4 questions to be addressed.

**Details Of Ethics Concerns:**

None.

---

> ### Author Response · Authors · 2023-11-13
> **Response to Reviewer Hczn**
>
> Thank you for the thoughtful consideration of the paper and constructive feedback. We’ve incorporated your feedback and uploaded a new revision of the paper.
>
> **W1: It is unclear to me why it is necessary to learn optimal Σ  via additional parameterization. What is the benefits of introducing additional degree of freedom here to learn Sigma? What if we fix Sigma without learning? Isn't it a simpler way to avoid degeneracy of matrix A? Please explain the motivation.**
>
> We compare fixed $\Sigma$ to learned $\Sigma$ in Figure 3. Like you mentioned, using a fixed $\Sigma$ indeed prevents degeneracy of $A$. However, learned $\Sigma$ works better empirically ($+3.5%$ compared to $+1.9%$, see Figure 3). Implementation wise, both approaches are very simple.
>
> The motivation to use a learned $\Sigma$ is to avoid having to perform an extensive grid search to find the optimal $\Sigma$ values. It is easier to let the model find the values itself.
>
> **W2: I understand that adding stochastic perturbation to position of the images makes sense in regularizing the model. However, why the same spectral decomposition is applied to features $s_x$ (by multiplying with $A$)? This step also lacks motivation and seems to be heuristic, please clarify on this point.**
>
> Without posing any constraint over $A$, we find that the weights of $A$ are quickly scaled down during training, setting $An=0$ to overcome the noise:
>
> 11: $n \sim \mathcal{N}(0, \sigma I)$
>
> 12: $m = An + \psi_{B_y} + \tilde{m}$
>
> Therefore, we resort to the basic MIM without stochasticity. To avoid this, we use $A$ to project both the noise tokens $n$ and the context tokens $s_x$:
>
> 11: $n \sim \mathcal{N}(0, \sigma I)$
>
> 12: $m = An + \psi_{B_y} + \tilde{m}$
>
> 13: $c = As_x + b + \psi_{B_x}$
>
> The motivation for using $A$ to project both the context features and noise can be understood by considering two extreme cases, when $A=0$, there is complete certainty about the positional embeddings of the masked tokens $\psi_{B_y}$ but all context is lost ($As_{x}=0$), thus making the MIM prediction task impossible. On the other hand, when $A$ is large the context information $As_{x}$ is preserved, but due to the large magnitude of $A$ the noise is amplified and camouflages the positional embedding features of the masked tokens: $An + \psi_{B_y}$, which makes the prediction task hard as well. This dual role of matrix $A$ forces the model to balance between location certainty and the influence of context features in predictions. It optimizes the trade-off for each feature, balancing their presence in predictions against the need for precise spatial locations.
>
> We discuss this in Section 3.1 (see “Avoiding a degenerate determinism solution”) and in Section 3.2 in the original submission but following the comment we revised the manuscript to make this more clear.
>
> **W3: What exactly architecture did the paper use to parameterize the matrix Σ? An architecture flow illustration will help better illustrate this mechanism. (figure 1 does not have this part ) Currently, I am not sure how the back-propagation of Σ  flows back to the network and how it affects the SSL learning with a positive gain.**
>
> We defined $\Sigma= \sigma AA^t$ (See revised manuscript Eq. 2) where $\sigma$ is a scalar hyperparameter and $A$ is a learned matrix.  However, instead of sampling from Eq.1 (where we cannot backprop through $A$), we use the reparametrization trick to sample noise $n \sim \mathcal{N}(0, \sigma I)$, and multiplying by $A$ to get the stochastic positional embeddings: $An + \psi_{B_y}$ (see revised manuscript Eq 3). This is differentiable w.r.t $A$ because the sampling distribution does not depend on $A$. Note that $Cov(An + \psi_{B_y}) = \Sigma$.
>
> We followed your suggestion and revised the architecture figure (Figure 2) to include the reparameterization trick to make it more clear (see new paper revision).
>
> **W4: I am not sure of the significance of proposition 1. I do not see why using this optimal predictor can help achieve better generalization ability of the SSL pretraining on downstream tasks.**
>
> The main goal of Proposition 1 is to provide insight to what is learned with StoP in a simple setting (one input and one output). In this case, we show that the optimal predictor explicitly models location uncertainty by performing spatial smoothing. We do not claim this property leads to better generalization (although we do see empirical downstream gains).
>
> To summarize, we think Proposition 1 provides a nice further analysis, but we are open to moving this into the appendix.

---

> > ### Author Response · Authors · 2023-11-19
> > **Followup**
> >
> > Dear reviewer, towards the end of the discussion phase, we trust that our response has successfully addressed your inquiries. We look forward to receiving your feedback regarding whether our reply sufficiently resolves any concerns you may have, or if further clarification is needed.

---

> ### Comment · Reviewer_Hczn · 2023-11-22
> **Thanks for your response**
>
> Thanks for the response! After reading the rebuttal, I think some of my concerns are addressed (empirical evidence showing the benefits of using learned Sigma vs the fixed Sigma). However, in terms of reusing the matrix A to $s_x$, I am still not convinced (W2). The current version of doing this projection lacks clear motivation and thus leaving it hard to judge the correctiveness. In this regard, I am afraid I agree with reviewer gTwQ, and I look forward to a better justification of the formulation.

---

> ### Author Response · Authors · 2023-11-22
>
> Dear reviewer, thank you for the reply and we are happy that some of your previous concerns are resolved.
>
> > However, in terms of reusing the matrix $A$ to $s_x$, I am still not convinced (W2). The current version of doing this projection lacks clear motivation and thus leaving it hard to judge the correctiveness.
>
> The reason for applying matrix $A$ to $s_x$ is to prevent the stochastic positional embeddings from collapsing into deterministic positional embeddings. Let's begin by describing why this collapse phenomenon happens, and subsequently, we will outline how the use of $A$ with $s_x$ provides an effective solution to address it.
>
> ## Stochastic positional embeddings collapse to deterministic
>
> By using the reparametrization trick, we generate stochastic positions as follows: $\hat{\psi}_i = An_i + \psi_i$ (Eq 3).
>
> It's important to note that $A$ regulates the noise level, and this noise disrupts the positional embeddings of the masked tokens. Therefore, better MIM predictions may be achieved without the presence of noise. Consequently, during training there is a risk of collapse into deterministic positional embeddings by setting $A=0$.
>
> Our experimental results confirm this. Without introducing a mechanism to prevent collapse, the empirical results resemble those obtained using deterministic Sine-Cosine features (see, for example, Table 6, under "Sine Cosine").
>
> ## Preventing Collapse
>
> Hence, in order to effectively capture location uncertainty through stochastic positional embeddings, it is crucial to prevent the occurrence of this collapse. While there might exist other ideas to address this issue, we employ a simple yet effective approach. This approach stands out as it doesn't necessitate additional losses, hyperparameters, or even learned weights.
>
> The idea is to use the matrix $A$ to project both $n$ and $s_x$. It's worth noting that in MIM models, there is a linear projection of $s_x$ from the encoder's dimension to the predictor's dimension and we replace it with $A$. For a detailed view of the differences between StoP and MIM, please refer to the revised paper, Algorithm 1.
>
> ## How does reusing $A$ to both $s_x$ and $n$ prevents collapse while promoting the modeling of location uncertainty?
>
> Using $A$ to project $s_x$ serves as a preventive measure against setting $A=0$, as doing so would eliminate crucial context information, making the MIM prediction task impossible. Or as pointed out by reviewer gTwQ: "StoP effectively prevents $A=0$ as it will lead to $As_x=0$".
>
> However, the model has to learn a matrix A that doesn't excessively amplify $s_x$ as this would result in amplifying the noise $n$ as well. Excessive amplification of the noise would camouflage the positional embeddings of the masked tokens, making their location very uncertain.
>
> ## Summary
>
> To summarize, without introducing a mechanism to prevent collapse, the positional embeddings become deterministic. We proposed to mitigate that by applying the same matrix $A$  both to the noise $n$ and to $s_x$. To learn a good $A$, the model has to trade off the importance of input context and the certainty in the masked tokens location.
>
> Please let us know if there are any other concerns, and we are open to hear more feedback and provide further clarification if needed. We discuss this topic at length in the recent revision (Section 3.2: “Avoiding a degenerate deterministic solution”).

---

> > ### Author Response · Authors · 2023-11-23
> >
> > Dear reviewer, we would greatly appreciate it if you could review our new response. We believe that we have effectively addressed all of your previous concerns. We actively stand by for the last few hours of the discussion phase.

---

### Author Response · Authors · 2023-11-13
**General response**

We thank the reviewers for their insightful and positive comments. The reviewers mentioned that stochastic positional embeddings (StoP) is “novel” (gTwQ) and an “intuitive idea to prevent the model from overfitting” (Hczn). Furthermore, reviewer 92Mr mentioned that the authors’ idea to prevent degenerate solution of the covariance matrix is a “smart modelling design choice”. Lastly, all reviewers are satisfied with the experimental study. Specifically, they mentioned StoP “significantly enhances the state-of-the-art method, demonstrating a noticeable improvement” (Hczn), “show consistent improvements” (92Mr), and that the experiments are “sufficient to support the proposed method” (gTwQ).

We addressed all the reviewers’ comments and incorporated their feedback to the new paper revision (new text highlighted in red).

---

### Meta-Review · Area_Chair_EtPy · 2023-12-12

**Metareview:**

The reviewers maintain concerns regarding the approach, despite the author rebuttal. The paper needs to be improved in its clarity of presentation before acceptance. In particular, the use of the same matrix A to project noise tokens and context tokens was not well justified in the submission. The authors provided further intuition in their responses, but neither of the two reviewers who raised the same concern were satisfied by the further justification in the author responses. For this reason, the authors are urged to solidify their justification of this choice in their method and potentially also explore alternative choices as they outline in one of their responses.

**Justification For Why Not Higher Score:**

Two reviewers raised major concerns about the methodology, despite author responses.

**Justification For Why Not Lower Score:**

N/A

---

### Decision · Program_Chairs · 2024-01-16

Reject